# Inversion of the Sign of the Short-Range Order as a Function of the Composition of Fe–Cr Alloys at Warm Severe Plastic Deformation and Electron Irradiation

**Kirill Kozlov** *[ID], **Valery Shabashov, Andrey Zamatovskii, Evgenii Novikov and Yurii Ustyugov**

Mikheev Institute of Metal Physics, Ural Branch, Russian Academy of Sciences, 620108 Ekaterinburg, Russia; shabashov@imp.uran.ru (V.S.); zamatovsky@imp.uran.ru (A.Z.); evg_nov@mail.ru (E.N.); ustyugov@imp.uran.ru (Y.U.)

* Correspondence: kozlov@imp.uran.ru

**Abstract:** This paper presents the results of a Mössbauer spectroscopy investigation of the processes in the binary alloys $Fe_{100-c}Cr_c$ ($c$, at. % = 6.0, 9.4, 13.2) and of the short-range (SR) atomic ordering accelerated by applying warm severe plastic deformation via high pressure torsion (HPT). After warm HPT treatment, in the vicinity of the concentration $c = 9$ at. %, there was revealed to be an inversion of the sign of the SR order, the anomaly of the formation of a Fe–Cr solid solution, which was predicted ab initio and is observed at long-term anneals and exposures to irradiation by electrons. The acceleration of the SR ordering at HPT is due to the continuous generation and a large number density of mobile point defects.

**Keywords:** Fe–Cr alloys; severe plastic deformation; electron irradiation; short-range ordering; point defects; Mössbauer spectroscopy

## 1. Introduction

Iron-rich Fe–Cr alloys are the base of conventional stainless steels, which are widely used in industry, including the nuclear industry, owing to their excellent mechanical and chemical properties and high resistance to void swelling [1–3]. To analyze the mechanisms of influence of high-energy particles on materials, data on the effect of large (megaplastic) deformation on the alloy structure can be used [4–8]. This is partly explained by the general determining factor of intense actions, i.e., the saturation of the structure with point defects. A specific feature of the Fe–Cr system is the miscibility gap taking place above ~9 at. % Cr [2,3]. Because of the spinodal decomposition of a homogeneous bcc Fe–Cr solid solution, the oversaturated high-chromium alloys and steels demonstrate a tendency to phase separation into Fe-rich ($\alpha$-phase) and Cr-rich ($\alpha'$-phase) fractions leading to the embrittlement found first upon ageing at 475 °C and named as 475 °C embrittlement. Besides, it is known [9–14] that an inversion in the type of short-range (SR) order formation takes place around 9–10 at. % Cr in the limits: from SR ordering, i.e., a tendency of atoms of different kinds neighbouring each other, to SR clustering, i.e., the opposite tendency of atoms of the same kind to neighbour each other.

It is evident that the nucleation of $\alpha'$ precipitates can occur only if the neighborhood of a Cr atom with the other Cr atoms is energetically preferable. In terms of the SR order formation, this means that the Fe–Cr alloys prone to $\alpha'$-precipitation are expected to demonstrate a tendency to SR clustering. On the other hand, it is known that, below 9 at. % Cr, the correspondent Fe–Cr alloys demonstrate a tendency to increase in terms of the number of neighborhoods between atoms of different kinds, namely Fe and Cr [9–14]. In [12–14], the inversion of the sign of the SR order is explained by the

specifics of the ferromagnetic iron matrix and by the specifics of an influence on this order from antiferromagnetic chromium, in other words: this inversion is explained by the magnetic nature of Fe–Cr alloys. Thus, the transition from unsaturated to oversaturated Fe–Cr alloys is expected to inevitably lead to a change in the type of SR order formation. Therefore, the inversion of the sign of SR ordering is most probably related to the limited Cr solubility in bcc iron and was predicted in ab initio theoretical calculations [12–14].

The sign inversion was first detected by means of small-angle neutron scattering and electrical resistivity measurements during thermal annealing above 700 K [10]. Then, it was confirmed by means of Mössbauer measurements in the Fe–Cr alloys irradiated by electrons at 420 K when irradiation-induced non-equilibrium defects accelerate the process of SR order formation at low temperatures [11]. Earlier, we found that severe plastic deformation (via HPT and ball milling) could also enhance the process of deformation-induced SR order formation in Fe–Mn and Fe–Cr alloys [4,5,15] at sufficiently low temperatures 370–570 K. In particular [4,5], we have observed that severe plastic deformation enhances SR ordering in $Fe_{86.8}Cr_{13.2}$ alloy. It is interesting to investigate in the next step whether the inversion of the type of SR formation also takes place in Fe–Cr alloys during severe plastic deformation.

## 2. Materials and Methods

The binary $Fe_{100-c}Cr_c$ alloys with bcc crystal lattice, of the following composition: $c = 6.0, 9.4, 13.2$ at. %, were prepared by melting in an induction furnace in argon atmosphere. The ingots (of 40 g) underwent annealing at 1073 K for 4 h in vacuum in the quartz ampoule, before being cooled in air after annealing. To this state of the material, we further in the text will refer to as-received. After annealing, the ingots obtained were spark cut into the plates ($7 \times 7$ mm$^2$ in size) and polished further mechanically and electrochemically down to a thickness of about 300 μm.

Samples blanked from the plates were deformed by HPT in the rotating Bridgman anvils with the degree $\varepsilon = 5.9$ ($n = 3$ revolutions of anvils) [16] and rate $2.4 \times 10^{-2}$ s$^{-1}$ ($\omega = 0.3$ rev/min) at temperatures of 80 and 573 K.

The scheme and procedure of the experiment is shown in [5]. After HPT, samples had a shape of a disc of 7 mm in diameter and 150 μm in thickness. For Mössbauer measurements, the disc was thinned, first mechanically with diamond paste and then electrochemically down to 20 μm. For the Mössbauer measurements, a full surface area of the sample was employed.

For the sake of a comparison of the deformation and irradiation effects, the 20 μm foils were irradiated by 5-MeV electrons at 420 K up to a fluence of $2 \times 10^{22}$ m$^{-2}$ [5,11].

Mössbauer γ-quantum absorption spectra on $^{57}$Fe were obtained at room temperature in the mode of constant acceleration with the $^{57}$Co(Cr) source. As a calibration standard, an iron foil was used. The measurements were carried out on the 20 μm foils and powder samples.

A package of special programs [17] was used for the modeling and approximation of experimental Mössbauer spectra by a sum of several sub-spectra, sextets $S(N_1, N_2)$, either of which corresponded to the various possible distributions of Cr atoms in the nearest coordination shells (CSs) of $^{57}$Fe atoms referred further to as the atomic configurations $(N_1, N_2)$. $N_1$ and $N_2$ are the numbers of chromium atoms in the first and second CSs, respectively. The integral intensity of each sextet $S(N_1, N_2)$ is proportional to the probability $W(N_1, N_2)$.

The redistribution of atoms during SR order formation was analyzed by means of Mössbauer spectra in the same way as it was done in [11,18–22], i.e., considering various possible distributions of Cr atoms in the two nearest coordination shells (CSs) of $^{57}$Fe atoms.

Separation (deconvolution) of partial contributions of the sextets $S(N_1, N_2)$ allows determining the effective chromium concentration $<c_{1,2}>$, i.e., averaged over two CSs, which can be represented in the following form:

$$< c_{1,2} > = \frac{Z_1 \cdot < c_1 > + Z_2 \cdot < c_2 >}{Z_1 + Z_2} \tag{1}$$

where $<c_1>$ and $<c_2>$ are the effective chromium concentrations over the first and the second CS, while $Z_1$ (=8) and $Z_2$ (=6) are the coordination numbers.

Knowledge of the value of $<c_{1,2}>$ and its comparison with the value of the impurity concentration at random distribution, $<c_{1,2}>_{random}$, permits one to determine the Cowley parameter:

$$\alpha_{1,2} = 1 - \frac{<c_{1,2}>}{<c_{1,2}>_{random}} \qquad (2)$$

In the case of a random distribution of atoms over the lattice sites, the value of $<c_{1,2}>$ is equal to $<c_{1,2}>_{random}$ of/in the alloy and we have $\alpha_{1,2} = 0$. In the case of the SR ordering with a tendency to neighboring between different atoms, $<c_{1,2}>$ is larger then $<c_{1,2}>_{random}$ and we have $\alpha_{1,2} < 0$ –further we'll refer to this case as to the case of SR ordering, for brevity, SRO (SR ordering). In the other case of the SR ordering with a tendency to neighboring between similar atoms, $<c_{1,2}>$ is less then $<c_{1,2}>_{random}$ and we have $\alpha_{1,2} > 0$. Further, we'll refer to this case as to the case of SR clustering or, for brevity, SRC (SR clustering).

## 3. Experimental Results

### 3.1. The Initial (As-Received) Condition of the Alloys

Figures 1–3 present Mössbauer spectra taken on the alloys under investigation. The result of calculating spectra of the alloys $Fe_{94}Cr_6$ and $Fe_{86.8}Cr_{13.2}$ in the initial condition has revealed a considerable difference between the values of $S(N_1,N_2)$ (which are proportional to the probability $W(N_1,N_2)$) characteristic of the initial (as-received) condition and the theoretical values of $S_{random}$ ($\sim W_{random}$) that were calculated for the alloys to be in the completely disordered states, see Figures 1 and 2 and Tables 1 and 2. The reason of this difference can be either a hardly controllable systematic inaccuracy (error) in determination of $S(N_1,N_2) \sim W(N_1,N_2)$, connected with the methodic specific features of Mössbauer spectroscopy, or the SR order in the initial (as-received) state. Due to the error of accounting for the sextets with small integral intensity in an experimental spectrum, only the sextets with a partial contribution exceeding 1% were taken into account in spectra approximation, i.e., $S(N_1,N_2) \geq 1\%$. The total contribution of these sextets, i.e., $\sum S(N_1,N_2)$, was about 95% (see Tables 1–3).

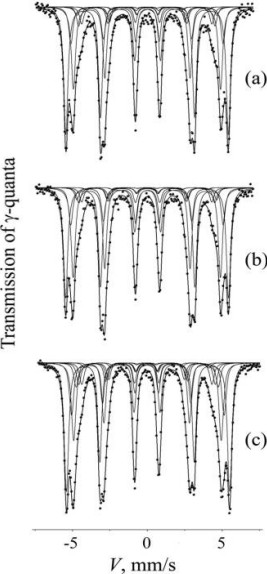

**Figure 1.** Spectra for the alloy $Fe_{94}Cr_6$. Treatment (state): (**a**) annealed at 1073 K (as-received); (**b**) as-received and irradiated with (fluence of) $F = 2 \times 10^{22}$ m$^{-2}$ at 420 K; (**c**) as-received and HPT ($\varepsilon = 5.9$, $n = 3$ rev.) at 573 K. Individual subspectra $S(N_1,N_2)$ corresponding to atomic configurations taken into account are indicated.

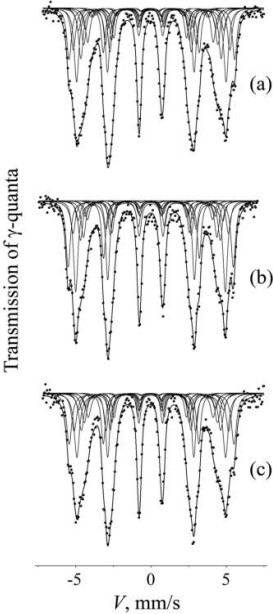

**Figure 2.** Spectra for the alloy $Fe_{86.8.2}Cr_{13.2}$. Treatment (state): (**a**) annealed at 1073 K (as-received); (**b**) as-received and irradiated with (fluence of) $F = 2 \times 10^{22}$ m$^{-2}$ at 420 K; (**c**) as-received and HPT ($\varepsilon = 5.9$, $n = 3$ rev.) at 573 K. Individual subspectra $S(N_1, N_2)$ corresponding to atomic configurations taken into account are indicated.

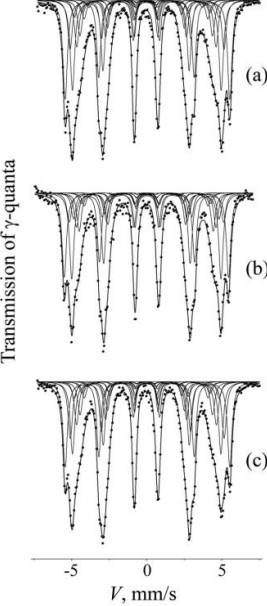

**Figure 3.** Spectra for the alloy $Fe_{90.6}Cr_{9.4}$. Treatment (state): (**a**) annealed at 1073 K (as-received); (**b**) as-received and irradiated with (fluence of) $F = 2 \times 10^{22}$ m$^{-2}$ at 420 K; (**c**) as-received and HPT ($\varepsilon = 5.9$, $n = 3$ rev.) at 573 K. Individual subspectra $S(N_1, N_2)$ corresponding to atomic configurations taken into account are indicated.

**Table 1.** Hyperfine Mössbauer parameters, intensities $S(N_1,N_2)$ of subspectra and $<c_{1,2}>$ for the $Fe_{94}Cr_6$ alloy in different states: $S_{random}$–theoretical; $S_{as\text{-}rec.}$–as received; $S_{irr.}$–irradiated; $S_{HPT\,(573)}$–HPT at 573 K; $S_{aged}$–aged at 773 K, 50 h; $S_{HPT\,(80)}$–aged and HPT at 80 K.

| Configuration | Isomer Shift | Hyperfine Magnetic Field | Sub-Spectrum Intensity $S(N_1,N_2) \sim W(N_1,N_2)$, % (±1) | | | | | |
|---|---|---|---|---|---|---|---|---|
| $(N_1,N_2)$ | $Is$, mm/s (±0.010) | $H$, kOe (±1) | $S_{random}$ | $S_{as\text{-}rec.}$ | $S_{irr.}$ | $S_{HPT\,(573)}$ | $S_{aged}$ | $S_{HPT\,(80)}$ |
| 0.0 | 0.015 | 338 | 42.0 | 40 | 38 | 38 | 38 | 41 |
| 0.1 | −0.006 | 316 | 16.1 | 17 | 18 | 16 | 18 | 17 |
| 1.0 | −0.020 | 305 | 21.5 | 23 | 24 | 24 | 24 | 23 |
| 0.2 | −0.029 | 294 | 2.7 | 3 | 3 | 5 | 3 | 3 |
| 1.1 | −0.039 | 283 | 8.2 | 9 | 9 | 9 | 9 | 8 |
| 2.0 | −0.054 | 272 | 4.8 | 5 | 6 | 6 | 6 | 5 |
| 1.2 | −0.066 | 261 | 1.3 | 2 | 1 | 1 | 1 | 2 |
| 2.1 | −0.076 | 250 | 1.8 | 1 | 1 | 1 | 1 | 1 |
| $<c_{1,2}>$, at. % (±0.2) | | | 5.6 | 5.7 | 6.0 | 6.2 | 6.0 | 5.7 |

$G$ (line width) for as-received, irradiated and aged states–0.34 mm/s, for HPT–0.35 mm/s. Values of quadrupole splitting ($Qs$) are close to zero and are not shown. $S(N_1,N_2) < 1\%$ are not shown.

**Table 2.** Hyperfine Mössbauer parameters, intensities $S(N_1,N_2)$ of subspectra and $<c_{1,2}>$ for the $Fe_{86.8}Cr_{13.2}$ alloy in different states: $S_{random}$–theoretical; $S_{as\text{-}rec.}$–as received; $S_{irr.}$–irradiated; $S_{HPT\,(573)}$–HPT at 573 K; $S_{aged}$–aged at 773 K, 50 h; $S_{HPT\,(80)}$–aged and HPT at 80 K.

| Configuration | Isomer Shift | Hyperfine Magnetic Field | Sub-Spectrum Intensity $S(N_1,N_2) \sim W(N_1,N_2)$, % (±1) | | | | | |
|---|---|---|---|---|---|---|---|---|
| $(N_1,N_2)$ | $Is$, mm/s (±0.010) | $H$, kOe (±1) | $S_{random}$ | $S_{as\text{-}rec.}$ | $S_{irr.}$ | $S_{HPT\,(573)}$ | $S_{aged}$ | $S_{HPT\,(80)}$ |
| 0.0 | 0.017 | 341 | 13.8 | 14 | 16 | 17 | 17 | 13 |
| 0.1 | −0.002 | 322 | 12.6 | 12 | 7 | 13 | 7 | 11 |
| 1.0 | −0.016 | 308 | 16.8 | 21 | 28 | 23 | 25 | 20 |
| 0.2 | −0.024 | 300 | 4.8 | 6 | 6 | 6 | 6 | 5 |
| 1.1 | −0.034 | 289 | 15.3 | 14 | 13 | 13 | 15 | 15 |
| 2.0 | −0.049 | 276 | 8.9 | 10 | 10 | 10 | 10 | 9 |
| 1.2 | −0.053 | 269 | 5.8 | 6 | 5 | 5 | 5 | 6 |
| 2.1 | −0.065 | 258 | 8.1 | 10 | 9 | 8 | 9 | 12 |
| 3.0 | −0.076 | 244 | 2.7 | 2 | 3 | 2 | 3 | 3 |
| 2.2 | −0.085 | 236 | 3.1 | 3 | 2 | 2 | 2 | 3 |
| 3.1 | −0.096 | 224 | 2.5 | 2 | 1 | 1 | 1 | 3 |
| $<c_{1,2}>$, at. % (±0.5) | | | 12.0 | 11.9 | 10.8 | 10.8 | 11.4 | 12.5 |

$G$ (line width) for as-received, irradiated and aged states–0.34 mm/s, for HPT–0.35 mm/s. Values of quadrupole splitting ($Qs$) are close to zero and are not shown. $S(N_1,N_2) < 1\%$ are not shown.

**Table 3.** Hyperfine Mössbauer parameters, intensities $S(N_1,N_2)$ of subspectra and $<c_{1,2}>$ for the $Fe_{90.6}Cr_{9.4}$ alloy in different states: $S_{random}$–theoretical; $S_{as\text{-}rec.}$–as received; $S_{irr.}$–irradiated; $S_{HPT\,(573)}$–HPT at 573 K; $S_{aged}$–aged at 773 K, 50 h; $S_{HPT\,(80)}$–aged and HPT at 80 K.

| Configuration | Isomer Shift | Hyperfine Magnetic Field | Sub-Spectrum Intensity $S(N_1,N_2) \sim W(N_1,N_2)$, % (±1) | | | | | |
|---|---|---|---|---|---|---|---|---|
| $(N_1,N_2)$ | $Is$, mm/s (±0.010) | $H$, kOe (±1) | $S_{random}$ | $S_{as\text{-}rec.}$ | $S_{irr.}$ | $S_{HPT\,(573)}$ | $S_{aged}$ | $S_{HPT\,(80)}$ |
| 0.0 | 0.015 | 340 | 25.1 | 26 | 26 | 25 | 24 | 24 |
| 0.1 | −0.027 | 320 | 15.6 | 13 | 13 | 14 | 12 | 12 |
| 1.0 | −0.012 | 308 | 20.8 | 23 | 23 | 23 | 26 | 24 |
| 0.2 | −0.019 | 300 | 4.1 | 4 | 4 | 4 | 4 | 5 |
| 1.1 | −0.028 | 288 | 13.0 | 12 | 13 | 13 | 12 | 13 |
| 2.0 | −0.039 | 274 | 7.6 | 10 | 9 | 10 | 11 | 10 |
| 1.2 | −0.046 | 266 | 3.4 | 3 | 3 | 3 | 3 | 3 |
| 2.1 | −0.059 | 256 | 4.7 | 5 | 4 | 4 | 4 | 5 |
| 3.0 | −0.060 | 242 | 1.6 | 1 | 2 | 2 | 2 | 2 |
| 2.2 | −0.077 | 235 | 1.2 | 2 | 1 | 1 | 1 | 1 |
| 3.1 | −0.088 | 221 | 1.0 | 0 | 1 | 1 | 1 | 1 |
| $<c_{1,2}>$, at. % (±0.2) | | | 8.8 | 8.9 | 8.8 | 8.9 | 8.9 | 8.9 |

$G$ (line width) for as-received, irradiated and aged states–0.34 mm/s, for HPT–0.35 mm/s. Values of quadrupole splitting ($Qs$) are close to zero and are not shown. $S(N_1,N_2) < 1\%$ are not shown.

Earlier [5,15], with the example of the binary alloys $Fe_{100-c}Mn_c$ ($c$, at. % = 4.1, 6.8, 9.0, 9.9) with bcc crystal lattice, it was shown that HPT at the temperatures 80 and 298 K "destroy" initial stages of SR ordering in the as-received and the aged state, bringing the state of solid solution closer (and closer) to a disordered state. Besides, the destruction of SR order was shown in the fcc binary alloy $Fe_{65}Ni_{35}$ [6], as well as the destruction of long-range order in the form of the dissolution of the intermetallic γ′ phase was revealed in ageing Fe–Ni–*Me* (Ti, Al) alloys of Invar composition [5,23].

For an elucidation of the question on the SR ordering in the initial as-received state, there were additionally investigations of the alloys after HPT at 80 K in the initial and the state aged at 773 K for 50 h (see Figures 4–6). Results of HPT cold deformation of the alloys $Fe_{94}Cr_6$, $Fe_{86.8}Cr_{13.2}$, and $Fe_{90.6}Cr_{9.4}$ in the initial and the aged state bring closer to each other the values of $W(N_{1,2}) = \sum S(N_1,N_2)$ (where $N_{1,2} = N_1 + N_2 = 0, 1$, and so on) and $W_{random}$, (see [4,5], Tables 1–3, and Figure 7). In particular, this is seen on the example of the most intense (and calculated with the most accuracy) sextets $S(0) \sim W(0)$ and $S(1) \sim W(1)$. The condition of solid solution (SS) in the alloys under investigation, which is described by the probability $W(N_{1,2})$ and $<c_{1,2}>$ and $α_{1,2}$ (both averaged over two CSs) after HPT at 80 K, permits one to consider it (SS condition) as the state most disordered when judged in average over two CSs. This does not mean the absence of SR order regions separately within the first and the second CS [22]. Because of the restricted capabilities of Mössbauer method (MM), one has the right to consider (i) the result of HPT at 80 K as referential standard corresponding to a disordered state, and (ii) the difference between $W_{random}$ and $W(N_{1,2})$ as the upper limit estimation of MM accuracy. Taking into account the disordering action of cold deformation, the degree of SR ordering has been estimated in terms of the Cowley parameter $α_{1,2}$ via the relationship as

$$\alpha_{1,2} = 1 - \frac{<c_{1,2}>}{<c_{1,2}>_{80K}} \tag{3}$$

where $<c_{1,2}>_{80K}$ is the effective chromium concentration $<c_{1,2}>$, i.e., averaged over two CSs, as the result of/after HPT at 80 K.

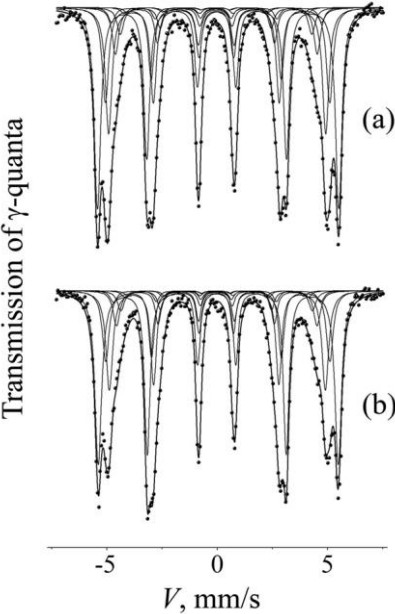

**Figure 4.** Spectra for the alloy Fe$_{94}$Cr$_6$. Treatment (state): (**a**) annealed at 1073 K (as-received) and aged at 773 K, 50 h; (**b**) aged and HPT ($\varepsilon$ = 5.9, *n* = 3 rev.) at 80 K.

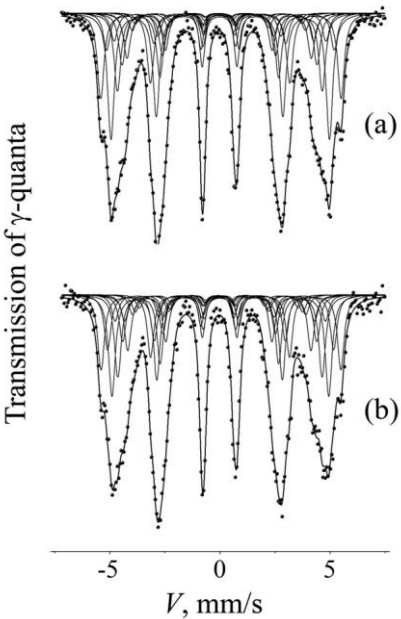

**Figure 5.** Spectra for the alloy Fe$_{86.8.2}$Cr$_{13.2}$. Treatment (state): (**a**) annealed at 1073 K (as-received) and aged at 773 K, 50 h; (**b**) aged and HPT ($\varepsilon$ = 5.9, *n* = 3 rev.) at 80 K.

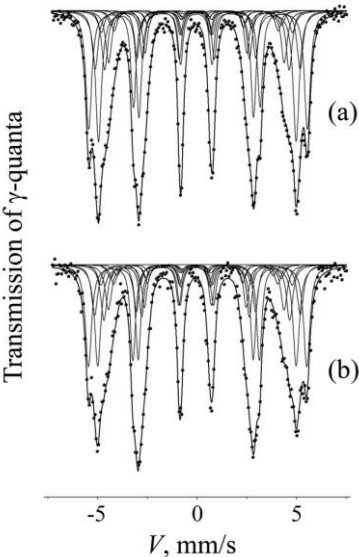

**Figure 6.** Spectra for the alloy Fe$_{90.6}$Cr$_{9.4}$. Treatment (state): (**a**) annealed at 1073 K (as-received) and aged at 773 K, 50 h; (**b**) aged and HPT ($\varepsilon$ = 5.9, *n* = 3 rev.) at 80 K.

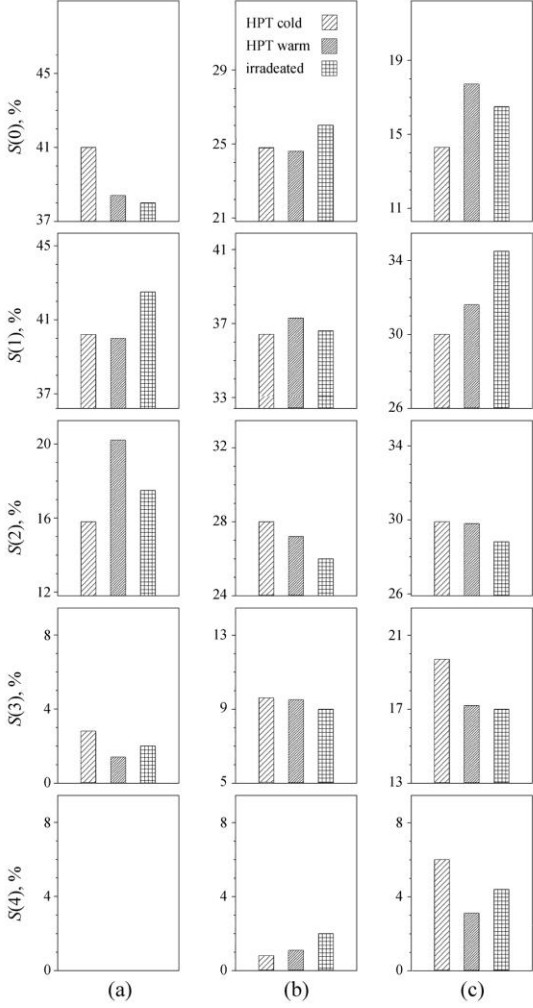

**Figure 7.** Bar histogram of the change of $S(N_{1,2}) \sim W(N_{1,2})$–of/(which is) the probability of occupation by the impurity Cr of the atomic configurations in the alloys: (**a**) Fe$_{94}$Cr$_6$; (**b**) Fe$_{90.6}$Cr$_{9.4}$; (**c**) Fe$_{86.8.2}$Cr$_{13.2}$ in dependence on treatment. Each treatment is indicated in the Figure.

### 3.2. Thermal Annealing and Irradiation of Fe–Cr Alloys by Electrons

Data on electron irradiation of the alloys under investigation agree well with the results from [10,11]. In the case of the alloy $Fe_{94}Cr_6$, after its annealing at 773 K (50 h) and irradiation with a fluence of $F = 2 \times 10^{22}$ m$^{-2}$ at 420 K one can observe changes in the probabilities $W(N_{1,2})$, which characterize the process of SR ordering, i.e., the deviation of $W(N_{1,2})$ for the as-received state from the value characteristic of the alloy in the disordered state (as a consequence of/after HPT at 80 K), see Table 1 and Figure 7a. The value of $<c_{1,2}>$ after electron irradiation (of the alloy) is equal to 6 at. %, i.e., it exceeds the effective concentrations (5.7 at. %) of the alloy in the as-received and the state deformed by HPT at 80 K. This indicates the presence of the process of SR ordering according to the type (called above by us) SRO with $\alpha_{1,2} = -0.053$, Figure 8a. In the alloy $Fe_{86.8}Cr_{13.2}$ after its irradiation, we have $<c_{1,2}> = 10.8$ at. %, i.e., this value is less than that of the effective concentration, 11.9 at. %, of the as-received condition and that (12.5 at. %) of the alloy deformed by HPT at 80 K, Table 2. This testifies to the realization of SR ordering by the type (called above by us) SRC with $\alpha_{1,2} = 0.136$, Figure 8c. In contrast to the alloys $Fe_{94}Cr_6$ and $Fe_{86.8}Cr_{13.2}$, in the alloy $Fe_{90.6}Cr_{9.4}$ under analogous action (impact) only little pronounced changes in $W(N_{1,2})$ and $<c_{1,2}>$ are revealed, Table 3 and Figure 8b.

In the spectrum, of the alloy $Fe_{94}Cr_6$, the sum $S(1) = S(0,1) + S(1,0)$ increases steadily, as well as does the sum $S(2) = S(0,2) + S(1,1) + S(2,0)$ – due to the increase in $S(0)$ and $S(N \geq 3)$. Obviously, the growth of $S(1) \sim W(1)$ and $S(1) \sim W(2)$ takes place as a consequence of the redistribution of chromium from the enriched configurations, which are represented by $S(N_{1,2} > 2) \sim W(N_{1,2} > 2)$ in $S(0) \sim W(0)$. Data of Table 1 and of the bar diagram (Figure 7a) show that the redistribution of chromium in the alloy $Fe_{94}Cr_6$ at warm (573 K) HPT is qualitatively analogous to the redistribution of chromium at the thermal and the irradiation effect and can be defined as SR ordering by the type SRO [11].

A result of warm 573K HPT of the alloy $Fe_{86.8}Cr_{13.2}$ is the growth of the intensity of the configurations $S(0) \sim W(0)$ and $S(1) \sim W(1)$ at the expense of decreasing $S(N_{1,2} \geq 2) \sim W(N_{1,2} \geq 2)$ originated from Cr-rich configurations, as shown in Table 2 and the bar histogram of Figure 7c. In this alloy, the value $<c_{1,2}>$ decreases in a way similar to that in the course of the annealing and irradiation by electrons in the near range of temperatures, which testifies to the realization of the processes of SR ordering by the type SRC [4,5], Figure 8c.

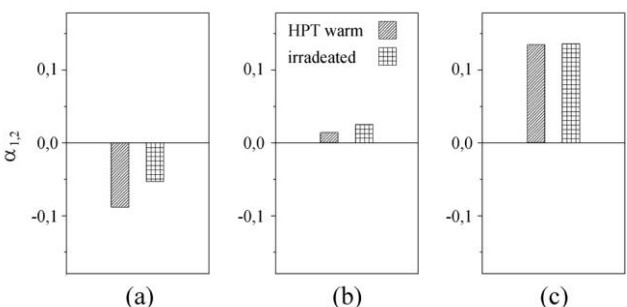

**Figure 8.** Cowley parameter $\alpha_{1,2}$ for (**a**) $Fe_{94}Cr_6$; (**b**) $Fe_{90.6}Cr_{9.4}$; (**c**) $Fe_{86.8.2}Cr_{13.2}$ alloys. Each treatment is indicated in the Figure.

### 3.3. HPT of Fe–Cr Alloys at 573 K

The results of the effect of the HPT at 573 K on the alloys under investigation are presented in Tables 1–3. Atomic redistribution is well illustrated by a bar diagram of changes in $S(N_{1,2})$, Figure 7.

A result of warm 573K HPT of the alloy $Fe_{86.8}Cr_{13.2}$ is the growth of the intensity of the configurations $S(0) \sim W(0)$ and $S(1) \sim W(1)$ at the expense of decreasing $S(N_{1,2} \geq 2) \sim W(N_{1,2} \geq 2)$ originated from Cr-rich configurations, as shown in Table 2 and the bar histogram of Figure 7c. In this alloy, the value $<c_{1,2}>$ decreases in a way similar to that in the course of the annealing and irradiation by electrons in the near range of temperatures, which testifies to the realization of the processes of SR ordering by the type SRC [4,5], Figure 8c.

An HPT at warm (573 K) deformation of the alloy $Fe_{90.6}Cr_{9.4}$ in the as-received condition, of an intermediate composition, slightly changes the intensities of $S(N_{1,2}) \sim W(N_{1,2})$ and $<c_{1,2}>$ (see Figure 3c and Table 3). This means that, in the vicinity of the range of compositions ~9 at. % Cr, the effect of warm HPT does not lead to the processes of SR ordering, while the thermal and irradiation effects do [10,11].

For the sake of a comparison of the deformation and irradiation effects, the 20 μm foils were irradiated by 5-MeV electrons at 420 K up to a fluence of $2 \times 10^{22}$ $m^{-2}$ [5,11].

## 4. Discussion

Thus, data on $W(N_{1,2})$, $<c_{1,2}>$, and $\alpha_{1,2}$ permit one to form an idea about the processes of the SR ordering at HPT and to compare it with the processes of the SR ordering activated in the alloys by the thermal and irradiation means of SR ordering initiation. Opposite changes in $S(N_{1,2})$ $\sim W(N_{1,2})$, $<c_{1,2}>$, and $\alpha_{1,2}$ (for one to observe) become the result of HPT at 80 K and 573 K in the alloys $Fe_{94}Cr_6$ and $Fe_{86.8}Cr_{13.2}$, both in the as-received and preliminary aged states, which testifies to the deformation-induced change of the sign of the SR order, Figures 7 and 8. In the alloy $Fe_{90.6}Cr_{9.4}$, warm HPT deformation little changes the values of $S(N_{1,2}) \sim W(N_{1,2})$, $<c_{1,2}>$, and $\alpha_{1,2}$. This alloy is allowed to be classified as one of those alloys that are characteristic of being close in composition to the concentration border of the change of the sign of SR order in conditions of applying warm HPT. Data on the cold deformation of the alloys $Fe_{94}Cr_6$ and $Fe_{86.8}Cr_{13.2}$ testify to either the incomplete homogenization in conditions of preliminary annealing at 1073 K for 4 h or realization of the initial stages of SR ordering as a result of insufficiently rapid cooling down to room temperature. The results of the effect of warm HPT on the $<c_{1,2}>$ and $\alpha_{1,2}$ of the alloys under investigation, in comparison with the effect of irradiation by electrons, testifies to realization of the processes of SR ordering, shown in Figures 7 and 8.

The observed ordering anomaly in the dilute Fe–Cr alloys with bcc crystal lattice is of magnetic origin. In the region of the composition less than 9 at. % and, in particular, in the alloy $Fe_{94}Cr_6$ under investigation, the process of SR ordering classified as of the type SRO is substantiated in the theoretical works [12,14]. According to theory, the sign inversion of SR order inversion (i.e., occurrence of an SRO region in the phase diagrams of dilute Fe–Cr alloys) is specific to the ferromagnetic state of the Fe matrix. In the very vicinity of the limiting solubility of Cr in Fe, the magnetic moments of the atoms of chromium impurity that have happened to become the nearest neighbors to one another are found to align in antiparallel manner to the magnetic moments of the atoms of the main constituent Fe. Such an addition of Cr to Fe to form the alloy entails that these chromium atoms exhibit their mutual antiferromagnetic exchange interaction (towards one another) due to what is called magnetic frustration. Therefore, Cr atoms tend to avoid each other on the nearest-neighbor shell, and thus to avoid magnetic frustration. Avoiding each other on the nearest-neighbor shell becomes impossible for the Cr atoms in concentrated Fe–Cr alloys, and the SR ordering switches to SRC type. In [13], it is clearly demonstrated that the pronounced ordering anomaly in the bcc dilute Fe–Cr alloys has a magnetic origin. From this it also follows that the disorder (thermal or composition-related) in the alignment distribution of the Fe magnetic moments is expected to strongly affect the effective cluster interactions by reducing the possibility for a Cr atom to align its magnetic moment antiparallel to the surrounding Fe atoms.

As has been already mentioned in previous studies [4,5,15], the mechanism of structure-phase transitions in conditions of severe plastic deformation at relatively moderate temperatures is explained by means of the concurrence between the processes of destruction of SR atomic order by moving dislocations, on the one hand, and the formation of the order by participation of mobile point defects, on the other. Thus, SR ordering after warm HPT is an accelerated process, since the SR order formation is completed in 10 min (while 3 revs of anvils) and is comparable (in the degree) with the results of (i) electron irradiation of fluence $2 \times 10^{22}$ $m^{-2}$ at 420 K for 6 h or (ii) thermal annealing at 673–773 K for 40–50 h. The prevailed contribution determining the efficiency of the ordering

in the range of deformations at temperatures above room temperature is connected with a high concentration of mobile vacancies and vacancy complexes, which are permanently generated in the course of the effect. The significance of the continuity of effect as the factor of the acceleration of SR ordering is confirmed by comparing the results of the warm HPT deformation and thermal anneals. In contrast to the SR ordering revealed in the alloys $Fe_{94}Cr_6$ and $Fe_{86.8}Cr_{13.2}$ after their warm HPT deformation, an isothermal annealing of the alloys preliminarily disordered by applying cold (298 K) HPT deformation at 573 K for 30 min does not lead to noticeable changes in the parameters of SR ordering. The factor of the continuity of the effect on the processes of both generating point defects and accelerating the diffusion was earlier demonstrated on the example of deformation-induced aging in the Fe-Ni-Al alloys capable of forming the intermetallics [23]. In particular, in the works on severe plastic deformation (by ECA—equal-channel angular pressing) with employment of the X-ray and positron annihilation, the authors have shown a realization of the increase in the concentration of vacancies and vacancy complexes up to the values typical of the state close to that of the melt [24,25]. With the increasing complexity of defects, which are vacancy complexes, at lowering temperature, we have an essential decrease in their efficiency in the processes of SR ordering. Since the concentration border of the change of the sign of SR ordering in Fe–Cr alloys of the composition in the vicinity of 9 at. % is stipulated by the effect of the composition and temperature on the magnetic state of the alloy [13,14], one has a right to consider that the sign of SR ordering at HPT is determined by the very same magnetic nature of the alloys. At the same time, the acceleration of the processes of SR ordering at HPT, which takes place similarly in the case of electron irradiation, is stipulated by the continuous generation and high concentration of the mobile point defects in the investigated range of the temperatures and compositions of Fe–Cr alloys [5].

Such a similarity between characteristic features of the acceleration of diffusion processes in the cases of application of the warm plastic deformation and irradiation by high-energy particles was demonstrated when studying the SR ordering in FCC Fe–Ni–*Me*(Ti, Al, Zr) alloys upon their intermetallic aging [5–8,23]. Thereby, it was shown (by us) that there is a common synergetic nature of the effect of the large plastic deformation (via rolling and HPT) and high-energy particles (such as electrons and fast neutrons). In both cases mentioned, the mechanism of action, the kinetics, and the final result of external action are determined by a vast number of the factors and, first of all, by the thermodynamic stimulus and mobility characteristics of the "participants" of diffusion. The complicated character and synergistic nature of transformations in conditions of large (severe) plastic deformation were shown in the theoretical works [26,27].

## 5. Conclusions

Using methods of Mössbauer spectroscopy, in the binary alloys $Fe_{100-c}Cr_c$ (*c*, at. % = 6.0, 9.4, 13.2), we have investigated the processes of SR ordering at warm (573 K) severe plastic deformation in the rotating Bridgman anvils. Lowering the temperature of deformation down to a cryogenic level (80 K) entails the destruction of the SR order, and thus affects the values of $<c_{1,2}>$, the so called effective concentration averaged over two (nearest) CSs, and of the parameter $\alpha_{1,2}$. Analysis of the distribution of atoms in conditions of warm (573 K) HPT showed that, in the vicinity (~9 at. %) of the concentration border, there takes place the change (inversion) of the sign of the parameter $\alpha_{1,2}$. This inversion was revealed earlier while annealing at 673–723 K and during irradiation by electrons [10,11]. The concentration anomaly of the change in the SR atomic order from the SRO type to the SRC type in the course of increasing Cr concentration is explained with the help of the existent specificity of the ferromagnetic state of the Fe matrix, namely, by the (event of) magnetic frustration [12–14], a phenomenon that leads to avoiding the occurrence of Cr atoms on the nearest-neighbor shell. In the alloy $Fe_{94}Cr_6$, one can observe SR ordering classified as SRO type, i.e., the effective concentration $<c_{1,2}>$ is greater than $<c_{1,2}>_{80K}$, whereas in the alloy $Fe_{86.8}Cr_{13.2}$, we have SR ordering classified as the SRC type, i.e., $<c_{1,2}>$ is less than $<c_{1,2}>_{80K}$. The established SR ordering is an example of accelerated ordering, and its degree depends on the temperature of deformation. A corollary

is made that at warm (573 K) deformation, the direction of ordering is determined via (and by the) composition of Fe–Cr alloys, a result similar to those we found in the cases of thermal annealing and electron irradiation at temperatures close to those of the thermal annealing. The acceleration of SR ordering in conditions of severe plastic deformation is explained by the saturation of the structure by mobile point defects.

**Author Contributions:** Conceptualization, V.S. and K.K.; methodology, V.S., E.N. and A.Z.; validation, V.S., K.K. and Y.U.; formal analysis, V.S., A.Z and K.K.; investigation, V.S., K.K.; writing—Original draft Preparation, K.K. and V.S.; writing—review and editing, V.S., K.K. and Y.U.; project administration, V.S.; funding acquisition, V.S. All authors have read and agreed to the published version of the manuscript.

**Funding:** The research was carried out within the state assignment of Ministry of Science and Higher Education of the Russian Federation (theme 'Structure' No. AAAA-A18-118020190116-6) supported in part by the Russian Foundation for Basic Research grant (project No. 18-03-00216).

**Conflicts of Interest:** The authors declare no conflict of interest.

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
