# Peer review of "Inversion of the Sign of the Short-Range Order as a Function of the Composition of Fe–Cr Alloys at Warm Severe Plastic Deformation and Electron Irradiation"

_metals, doi:10.3390/met10050659_

Round 1
Reviewer 1 Report
The inversion of SRO in annealed, HPT-deformed, and electron-irradiated Fe-Cr alloys is investigated using Mössbauer spectroscopy. Severe plastic deformation (SPD) at a low temperature of 80 K is found to result in a random distribution of Cr, while at an elevated temperature of 573 K SPD is found to accelerate the formation of SRO in the studied alloy specimens.The acceleration effect is ascribed to the increased concentration of point defects (vacancies) that are produced by SPD and are mobile at the elevated temperature.
The study has been performed at a high technical level. The results are original, their analysis employs a modern analytic technique based on deconvolution of the measured spectra using a modern software.
Therefore, the paper is worth being published. However: 1. The language of the paper needs additional editing to remove numerous errors that make reading the paper, in the current version, a difficult task. This mainly concerns Sections 4 (Discussion) and 5 (Conclusions) which are very unclearly written and should be re-written. Other points (scientific and technical) that require authors' attention are enumerated below. 2. The origin of SRO inversion is unclearly formulated in the paper. Thus, on Page 1 (lines 39-43) the inversion is attributed to exceeding the solubility limit of Cr in Fe, whereas on Page 11 (lines 272.273) it is attributed to the change in the magnetic state of the alloy.
According to theory, the SRO inversion (= occurrence of a SO region in dilute Fe-Cr alloys) is specific of the ferromagnetic state of the Fe matrix.
There any 2 Cr atoms tend to avoid each other on the nearest-neighbor shell, to avoid magnetic frustration (Cr moments are aligned antiparallel to Fe moments). Avoiding each other on the nearest-neighbor shell becomes impossible for Cr atoms in concentrated Fe-Cr alloys, and the SRO switches to SD type.
The inversion of SRO, as a function of Cr concentration, does not require exceeding the solubility limit of Cr nor any change in the magnetic state of the alloy. However, the compositions for some of the studied alloys correspond to the 2-phase region at 80 or 573 K, so that the bcc solid solutions there are metastable.
All these details should be carefully explained in the paper, to avoid confusion. 3. The authors introduce a joint SRO parameter that combines together the Cr-Cr pair correlations on the first and second coordination shells. The authors are kindly requested to explain the reasons for their choice of SRO parameter in the paper: Why is it better than using 2 separate SRO parameters? 4. Some suggestions to fix typographic/language errors: line 34: atoms of different origin --> atoms of different kind
line 35: an opposite tendency --> the opposite tendency
line 35: the same origin --> the same kind
line 38: typical of \alpha^\prime-precipitating --> prone to \alpha^\prime-precipitation
line 133: one has a right to --> one has the right to
lines 156, 158, and 165: HTP --> HPT
line 159: With taking into account --> Taking into account
line 328: 64 --> 164
Author Response
Dear Sir,
Thank you for your close peer-reviewing. It helped us exclude many mistakes.
The authors have taken into account all remarks and performed corrections, having marked these corrections in the text by a yellow color.
Critical remarks:
- The language of the paper needs additional editing to remove numerous errors that make reading the paper, in the current version, a difficult task. This mainly concerns Sections 4 (Discussion) and 5 (Conclusions) which are very unclearly written and should be re-written. Other points (scientific and technical) that require authors' attention are enumerated below.
– The answer:
The authors have performed the correction and additional editing of the text and, especially, of the Sections 4 (Discussion) and 5 (Conclusions). The part of the text that underwent the correction is marked by yellow.
- The origin of SRO inversion is unclearly formulated in the paper. Thus, on Page 1 (lines 39-43) the inversion is attributed to exceeding the solubility limit of Cr in Fe, whereas on Page 11 (lines 272.273) it is attributed to the change in the magnetic state of the alloy.
According to theory, the SRO inversion (= occurrence of a SO region in dilute Fe-Cr alloys) is specific of the ferromagnetic state of the Fe matrix. There any 2 Cr atoms tend to avoid each other on the nearest-neighbor shell, to avoid magnetic frustration (Cr moments are aligned antiparallel to Fe moments). Avoiding each other on the nearest-neighbor shell becomes impossible for Cr atoms in concentrated Fe-Cr alloys, and the SRO switches to SC type.
The inversion of SRO, as a function of Cr concentration, does not require exceeding the solubility limit of Cr nor any change in the magnetic state of the alloy. However, the compositions for some of the studied alloys correspond to the 2-phase region at 80 or 573 K, so that the bcc solid solutions there are metastable.
All these details should be carefully explained in the paper, to avoid confusion.
– The answer:
- In Section 1 (Introduction) the textual fragment that starts with (“On the other hand…”) is revised to: “On the other hand, it is known that below 9 at. % Cr the correspondent Fe–Cr alloys demonstrate a tendency to the increase in the number of neighborhoods between atoms of different kinds, namely, of Fe and Cr [9–14]. In [12–14] the inversion of the sign of the SR order is explained by the specifics of the ferromagnetic iron matrix and by the specifics of an influence on this order from antiferromagnetic chromium, in other words: this inversion is explained by the magnetic nature of Fe-Cr alloys”.
- In Section 4 (Discussion) a physical interpretation of the sign of SR ordering is given in accordance with the theoretical works of [12–14]: “The observed ordering anomaly in the dilute Fe–Cr alloys with bcc crystal lattice is of magnetic origin. In the region of the composition less than 9 at.% and, in particular, in the alloy Fe94Cr6 under investigation, the process of SR ordering by the type SRO is substantiated in the theoretical works [12–14]. According to theory, the sign inversion of SR order inversion (i.e., occurrence of a SRO region in [the phase diagrams of] dilute Fe–Cr alloys) is specific of the ferromagnetic state of the Fe matrix. In the very vicinity of the limiting solubility of Cr in Fe, the magnetic moments of the atoms of chromium impurity that have happened to become the nearest neighbors to one another are found to align in antiparallel manner to the magnetic moments of the atoms of the main constituent Fe. Such an addition of Cr to Fe to form the alloy entails that these chromium atoms exhibit their mutual antiferromagnetic exchange interaction (towards one another), which is a cause of what is called the magnetic frustration. Therefore, Cr atoms tend to avoid each other on the nearest-neighbor shell, and thus to avoid magnetic frustration. Avoiding each other on the nearest-neighbor shell becomes impossible for the Cr atoms in concentrated Fe-Cr alloys, and the SR ordering switches to SRC type. In [13] it is clearly demonstrated that the pronounced ordering anomaly in the bcc dilute Fe–Cr alloys has a magnetic origin. From this it also follows that the disorder (thermal or composition-related) in the alignment distribution of the Fe magnetic moments is expected to strongly affect the effective cluster interactions by reducing the possibility for a Cr atom to align its magnetic moment antiparallel to the surrounding Fe atoms.”.
- In Section 5 (Conclusions) the following correction is performed: “The concentration anomaly of the change in the short range atomic order from the SO type to the SC type in the course of increasing Cr concentration is explained with the help of the existent specificity of the ferromagnetic state of the Fe matrix, namely, – by the (event of) magnetic frustration [12–14] – a phenomenon that leads to avoiding the occurrence of Cr atoms on the nearest-neighbor shell.”
The procedure of preparing specimens included their annealing at 1073 K for 4 h with the aim of homogenizing in the one-phase (existence) field of the alloys of investigated compositions.
- The authors introduce a joint SRO parameter that combines together the Cr-Cr pair correlations on the first and second coordination shells. The authors are kindly requested to explain the reasons for their choice of SRO parameter in the paper: Why is it better than using 2 separate SRO parameters?
– The answer:
The aim of the work was in the establishment of the general pattern of sign inversion and concentration boundary, as well as SR ordering acceleration, in conditions of (i) warm sever plastic deformation, (ii) long-term annealing, and (iii) irradiation by high-energy electrons. The results permit making a conclusion on the increase in the concentration of mobile point defects in conditions of sever plastic deformation.
Our approach typical of using the combined Cr-Cr pair correlations on/over the first and second coordination shells is justified by the possibility of comparing the results of our work and those of Mössbauer works on the inversion of the sign of SR ordering in the course of annealing or irradiation (e.g., [11]). Fig.7 of the given (our) work allows one to evaluate (i) the unified nature of the change of the probability of occupation by the Cr of the atomic configuration and (ii) the presence of the concentration border of the inversion of the sign of SR ordering.
The reviewer is right in that knowledge of individual <c1> (a1) and <c2> (a2) may be of interest for clarification of the model (simulation) of SR ordering in the first and second CSs of Fe atom.
- Some suggestions to fix typographic/language errors:
line 34: atoms of different origin --> atoms of different kind
line 35: an opposite tendency --> the opposite tendency
line 35: the same origin --> the same kind
line 38: typical of \alpha^\prime-precipitating --> prone to \alpha^\prime-precipitation
line 133: one has a right to --> one has the right to
lines 156, 158, and 165: HTP --> HPT
line 159: With taking into account --> Taking into account
line 328: 64 --> 164
– The answer:
We accepted all the corrections proposed by the reviewer.
Yours truly,
the authors

Reviewer 2 Report
Authors investigated the presence of HPT-induced SRO (resulting from participation of mobile point defects) in Fe-Cr alloys. Besides the thermally induced SRO or SRC, deformation-induced SRO is a quite important feature for controlling mechanical properties. Some minor issues remain.
In introduction,
- authors mentioned "Because of the limited Cr solubility, ~~~~ 475C embrittlement.". --> Rather than addressing the embrittlement, it is better important for addressing spinodal decomposition in text, because this article deals with SRO and SRC as well.
- Authors used SRO or SR ordering in text. Please unify it as one term either SRO or SR ordering in the manuscript.
- At the end of Introduction section, authors mentioned "severe plastic deformation could enhance SRC". Please clarify ir as deformation-induced SRC. This term is useful for differentiating the thermally induced SRC or SRO. As like, this case applies to SRO case, which is referred to as deformation-induced SRO in Fe-Cr alloys.
- On line 121, grammar check !!, is it SRO or long-range order ?
- On line 235, authors point out again which figures in this article revealing the mechanism SRO from mobile point defects.
- On line 281, what is the saturation of the structure by the mobile point defects ? Please explain it concicely.
Author Response
Dear Sir,
We thank you for your critical remarks.
The authors have taken into account all remarks and performed corrections, having marked these corrections in the text by a yellow color.
Critical remarks:
- Authors mentioned "Because of the limited Cr solubility, ~ 475 C embrittlement.". --> Rather than addressing the embrittlement, it is better important for addressing spinodal decomposition in text, because this article deals with SRO and SRC as well.
– The answer:
The critical remark no.1 has been taken into account: in the Introduction the following phrase was inserted: “Because of spinodal decomposition of a homogeneous bcc Fe-Cr solid solution, the oversaturated high-chromium alloys and steels demonstrate a tendency to phase separation into Fe-rich (a-phase) and Cr-rich (a¢-phase) fractions leading to the embrittlement found first upon ageing at 475°C and named as 475°C embrittlement”.
- Authors used SRO or SR ordering in text. Please unify it as one term either SRO or SR ordering in the manuscript.
– The answer:
The critical remark no.2 has been taken into account: correspondent corrections were made all over the text, namely, the term SR ordering is used. In dependence of the sense (sign) of the SR ordering we have introduced the abbreviation SRO – for a1,2 < 0, and the abbreviation SRC – for a1,2 > 0; correction (clarification) of the terminology is added in the final block paragraph of the Section 2 (Materials and methods).
- At the end of Introduction section, authors mentioned "severe plastic deformation could enhance SRC". Please clarify it as deformation-induced SRC. This term is useful for differentiating the thermally induced SRC or SRO. As like, this case applies to SRO case, which is referred to as deformation-induced SRO in Fe-Cr alloys.
– The answer:
In Section 1 (Introduction) we have made an additional textual fragment on the nature of deformation-induced SR ordering.
- On line 121, grammar check !!, is it SRO or long-range order ?
– The answer:
The grammatical error is eliminated.
- On line 235, authors point out again which figures in this article revealing the mechanism SRO from mobile point defects.
– The answer:
We added to the text an info on the comparison between the warm deformation and isothermal annealing of preliminarily deformed samples: “The prevailed contribution determining the efficiency of the ordering in the range of deformations at temperatures above room temperature, – is connected with a high concentration of mobile vacancies and vacancy complexes, which are permanently generated in the course of the effect. The significance of the continuity of effect as the factor of the acceleration of short range ordering is confirmed by comparing the results of the warm HPT deformation and thermal anneals. In contrast to the short range ordering revealed in the alloys Fe94Cr6 and Fe86.8Cr13.2 after their warm HPT deformation, an isothermal annealing of the alloys preliminarily disordered by applying cold (298 K) HPT deformation at 573 К for 30 min does not lead to noticeable changes in the parameters of SR ordering. The factor of the continuity of the effect on the processes of both generating point defects and accelerating the diffusion, – was earlier demonstrated on the example of deformation-induced aging in the Fe-Ni-Al alloys capable of forming the intermetallics [23]”.
- On line 281, what is the saturation of the structure by the mobile point defects? Please explain it concicely.
– The answer:
For the explanation, in the section Discussion we added info on the significance of the continuity of the effect that causes generation and saturation of the structure by the vacancies and vacancy complexes (see additions on the critical note no.5).
With best regards,
the authors
